# Inverse Modeling of Seepage Parameters Based on an Improved Gray Wolf Optimizer

Yongkang Shu [1], Zhenzhong Shen [1,2], Liqun Xu [1,*], Junrong Duan [1], Luyi Ju [1] and Qi Liu [3]

1   College of Water Conservancy and Hydropower Engineering, Hohai University, Nanjing 210098, China
2   State Key Laboratory of Hydrology-Water Resources and Hydraulic Engineering, Hohai University, Nanjing 210098, China
3   Datang Hydropower Science & Technology Research Institute Co., Ltd., Nanning 530007, China
*   Correspondence: xlq@hhu.edu.cn

**Abstract:** The seepage parameters of the dam body and dam foundation are difficult to determine accurately and quickly. Based on the inverse analysis, a Gray Wolf Optimizer (GWO) was introduced into this study to search the target hydraulic conductivity. A novel approach for initialization, a polynomial-based nonlinear convergence factor, and weighting factors based on Euclidean norms and hierarchy were applied to improve GWO. The practicability and effectiveness of Improved Gray Wolf Optimizer (IGWO) were evaluated by numerical experiments. Taking Kakiwa dam located on the Muli River of China as a case, an inversion analysis for seepage parameters was accomplished by adopting the proposed optimization algorithm. The simulated hydraulic heads and seepage volume agree with measurements obtained from piezometers and measuring weir. The steady seepage field of the dam was analyzed. The results indicate the feasibility of IGWO in determining the seepage parameters of Kakiwa dam.

**Keywords:** inverse analysis; hydraulic conductivities; Gray Wolf Optimizer

## 1. Introduction

In hydraulic engineering, seepage parameters of dams and dam foundations change with operating time and loading conditions. The changes in seepage parameters weaken the strength of the structure and lead to failure. Seepage analysis is commonly used to monitor the working conditions of dams and dam foundations for the safety of hydraulic projects [1–5]. The hydraulic conductivity, a key parameter in seepage analysis, is closely related to the accuracy of the analysis results. Minimized error between the simulated and actual values of hydraulic conductivity could improve the reliability of the analysis. In-situ tests have been proven to be helpful in determining the hydraulic conductivity of dams and dam foundations. The hydraulic conductivity determined by in-situ testing agrees with the actual value when the test samples are small. However, this method is time-consuming and costly when there are large quantities of models. Another method to solve this problem is inverse analysis. The inverse research based on monitoring data and numerical simulation results demonstrates economy and efficiency. The essence of the inverse analysis is to determine the hydraulic conductivity by measurements and simulated results. In the inversion analysis, optimization algorithms are widely applied for iterative search over a range of hydraulic conductivity values. The optimal hydraulic conductivity is determined by iteration while minimizing the objective function.

Considering the repetition of the iterative process, optimization algorithms are widely used to improve efficiency and accuracy in the searching process of the inverse problem. For example, on the basis of the Radial Basis Function (RBF) neural network optimized by Particle Swarm Optimization (PSO), Chi et al. [6] constructed an inverse model for the permeability coefficient of a high core rockfill dam; Combining error Back-Propagation Neural Network (BPNN) and Genetic Algorithm (GA), Deng and Lee [7] proposed an

inverse analysis method for determining the displacements. This method was successfully applied in the displacement identification of the lock profile of the Three Gorges Project, which led to reasonable results. Zhao et al. [8] developed the differential evolution (DE) algorithm to determine soil parameters in the field of deep excavation, which improved the stability of the backtracking parameters. Simulated annealing [9–12] and ant colony optimization [13–17] have also been extensively used in the inverse problem. Significantly, much progress has been made in the research field of seepage because of optimization algorithms [18–25]. Tan et al. [26] proposed a biological immune mechanism-based quantum particle swarm optimization (IQPSO) algorithm to solve the inversion problem of seepage parameters. Based on back propagation neural network (BPNN) and genetic algorithm (GA), Zhou et al. [27] developed a new approach for inverse modeling of the transient groundwater flow in dam foundations, which improved the uniqueness and reliability of the inversed results and made tractable the large-scale inverse problems in engineering practices. Zhang et al. [28] proposed an inverse analysis model by using the genetic algorithm (GA) and finite element analysis technology, to solve the calcium leaching problems.

Although optimization algorithms are frequently employed for inverse problems, they suffer from low accuracy, slow convergence, and poor robustness. The Gray Wolf Optimizer (GWO) proposed by Mirjalili [29] has been shown to be efficient and intelligent in engineering optimization. Mirjalili [29] compared the performance of GWO with Particle Swarm Optimization (PSO), Gravity Search Algorithm (GSA), Differential Evolution (DE), Evolutionary Programming (EP), and Evolutionary strategy (ES). The results demonstrate that the GWO can provide very competitive results compared to these well-known metaheuristics. It has been extensively adopted in various fields due to its simple structure, fewer parameters, and easy coding implementation. However, it tends to converge to locally optimal solutions. In addition, suboptimal values could result from completely randomized initial populations. Therefore, strategies of improvement are proposed as necessary.

Generally, there are three main strategies to improve the GWO, including adjustments of initial populations, convergence factor, and formula of a location update [30]. Pradhan et al. [31] combined the concept of opposition with GWO, initially providing a uniform population for the algorithm. Long et al. [32] introduced the theory of good point set to population initialization, which improved the homogeneity of the population. Based on this, Long et al. [33] considered the dynamics of the iterative process and proposed an equation of the convergence factor based on the number of iterations. This exponential function simulates the iterative process and balances the local and global search to a certain extent. Mittal et al. [34] described the decay process of the parameter by an exponential function, which leads to improved accuracy for GWO. Salgotra et al. [35] applied the spiral property from the whale optimization algorithm (WOA) to the GWO, which solves the premature convergence in the evolutionary algorithm. Mostafa et al. [36] introduced variational operators to update the location of the individual in GWO and improved the algorithm's performance. Gupta and Deep [37] adopted a random wandering strategy to enhance the accuracy of the algorithm.

A great deal of research has been conducted on the improvement of GWO. However, local and global search, along with homogenization and randomization, cannot be relatively balanced by these improvements. GWO based on these strategies is still limited in terms of efficiency and accuracy. Therefore, there remains potential for improvement across these dimensions.

Based on the evolution of the GWO, three strategies are proposed to ensure the accuracy and efficiency of the algorithm. Initial populations of semi-uniform and semi-random were proposed for rational initialization. A polynomial-based nonlinear convergence factor was applied to maintain a balance between global and local search. Weighting factors based on Euclidean norms and hierarchy were given to dynamically update the wolves' positions for jumping out of the local optimum at the late stage of the iterative search [38,39]. The Improved Gray Wolf Optimizer (IGWO) has been proven to be effective through numerical

experiments. This algorithm was applied for the inversion model of the Kakiwa Dam located in Sichuan Province, China. The fitness function is constructed by measurements of piezometers and measuring weir. The finite element method was used to simulate the seepage process under the assumption of steady flow. The free tetrahedral grid is used to construct the two-dimensional mesh of the dam. The finite element calculation is carried out in COMSOL Multiphysics, while the iterative control and data extraction are implemented in MATLAB. The objective hydraulic conductivity and the corresponding fitness value were determined when the maximum number of iterations was achieved. The seepage field at the dam site was also presented.

## 2. Improved Gray Wolf Optimizer

### 2.1. Overview of Gray Wolf Optimizer

The Gray Wolf Optimizer (GWO) is a new group intelligence algorithm considering gray wolves' hierarchy and group hunting. There is a strict hierarchy in the gray wolf population. The population is classified into four levels of status in accordance with the fitness values of individuals. Wolves in the first level of the population are responsible for making decisions and leading the group in hunting. Wolves in the second level take responsibility for helping to manage the group. Wolves in the third level of the group obey the orders of the first two levels of gray wolves. All the remaining populations are set at the fourth level.

The hunting process tends to be taken as group action in gray wolf populations, which could be summarized by tracking, encircling, and attacking. The rank in the wolf pack changes dynamically with the individual fitness value in the hunting process. The fitness value can be considered the distance between a wolf and its prey. This means that the closer the distance to the target, the higher the level of the wolf.

Let $\alpha$, $\beta$, and $\gamma$ represent the three dominant wolves in rank order. The mathematical model of gray wolf hunting is established as described in Equation (1).

$$\begin{cases} d = \left| C \cdot X_p(t) - X(t) \right| \\ X(t+1) = X(t) - A \cdot d \end{cases} \tag{1}$$

where $d$ is the perceived distance between the gray wolf and the prey; $t$ represents the number of iterations. $C$ denotes the coefficient vector; $X_p(t)$ and $X(t)$ are the positions of the wolf and the prey, respectively; $X(t+1)$ means the position of the wolf after the iteration; $A$ stands for the coefficient vector.

The expressions for $A$ and $C$ are shown in Equation (2).

$$\begin{cases} A = a \cdot (2m_1 - 1) \\ \quad C = 2m_2 \end{cases} \tag{2}$$

where $a$ is the convergence factor; $m_1$ and $m_2$ are both random numbers between 0 and 1.

It is assumed that the first three levels of gray wolves have a better perception of the location of the prey. The populations in the fourth layer decide the direction and distance of the next movement according to the positions of the three dominant wolves. The wolf's position in the fourth rank is updated according to Equation (3).

$$X(t+1) = \frac{1}{3} \sum_{i=1}^{3} X_i \tag{3}$$

where $X(t+1)$ denotes the position of the wolf $\varphi$ after update; $X_i (i = 1, 2, 3)$ represent the position vectors of the wolf $\alpha$, $\beta$, and $\gamma$, respectively.

The position vectors of $\alpha$, $\beta$, and $\gamma$ are expressed by Equation (4).

$$\begin{cases} X_1 = |X_\alpha - A_1 \cdot d_1| \\ X_2 = |X_\beta - A_2 \cdot d_2| \\ X_3 = |X_\gamma - A_3 \cdot d_3| \end{cases} \tag{4}$$

where $X_i(i = 1, 2, 3)$ stand for the position vectors of the wolf $\alpha$, $\beta$, and $\gamma$, respectively; $X_j(j = \alpha, \beta, \gamma)$ represent the prey position perceived by $\alpha$, $\beta$, and $\gamma$, respectively. $A_i(i = 1, 2, 3)$ are the coefficient vectors; $d_i(i = 1, 2, 3)$ mean the distance between the three dominant wolves and the prey.

$d_i(i = 1, 2, 3)$ could be expressed as Equation (5) follows.

$$\begin{cases} d_1 = |C_1 \cdot X_\alpha - X(t)| \\ d_2 = |C_2 \cdot X_\beta - X(t)| \\ d_3 = |C_3 \cdot X_\gamma - X(t)| \end{cases} \tag{5}$$

where $C_i(i = 1, 2, 3)$ are the coefficient vectors; $X(t)$ is the wolf's position in the $t$-th iteration.

### 2.2. Strategies of Improvement

#### 2.2.1. Initial Populations of Semi-Uniform and Semi-Random

The instability of the solution could be increased by completely randomized initial populations, which leads to unstable results. A novel approach for initializing populations was presented to balance uniformity and randomness. The solution range is divided into intervals equidistantly according to the population size. The $j$-th interval can be expressed by Equation (6).

$$\Delta_j = \left[ lb + \frac{ub - lb}{S}(j-1), lb + \frac{ub - lb}{S}j \right] (j = 1, 2 \ldots, S) \tag{6}$$

where $\Delta_j$ is the $j$-th interval; $lb$ and $ub$ are the upper and lower bounds of the solution set, respectively; $S$ is the population size.

Generate a random initial solution in each interval to ensure randomness. The initial populations are uniformly distributed in the solution space without losing randomness. The expression of the initial solution is given in Equation (7).

$$R_j = lb + \frac{ub - lb}{S}j + rand() \cdot \frac{ub - lb}{S}(j = 1, 2 \ldots, S) \tag{7}$$

where $R_j$ is the initial solution of the $j$-th interval; $lb$ and $ub$ are the upper and lower bounds of the solution set, respectively; $S$ is the population size; $rand()$ is a random real number between 0 and 1.

#### 2.2.2. Polynomial-Based Nonlinear Convergence Factor

The way to search for prey is determined by the coefficient vector $A$. The gray wolf can be in any position between the current individual and the prey at the next moment when $|A| < 1$, indicating that the next position of the wolf will be closer to the location of the prey. This is considered a local search. When $|A| > 1$, the next location of the wolf will be further away from the prey than the current location. Gray wolves tend to search over a wider area, which is considered a global search. The positions of gray wolves change with a rapid convergence speed. In this case, the search step size of the gray wolf becomes smaller, thus achieving a refined search. $A$ varies dynamically with convergence factor $a$. Considering good symmetry and smoothness, a function based on a third-degree polynomial was used to fit the convergence factor $a$. The expression for $a$ is shown in Equation (8).

$$a(t) = r_1 \left(\frac{t}{T}\right)^3 + r_2 \left(\frac{t}{T}\right)^2 + r_3 \frac{t}{T} + r_4 \tag{8}$$

where $a$ is the convergence factor; $r_i(i = 1, 2, 3, 4)$ denote the real-valued parameters. $t$ represents the number of iterations. $T$ is the maximum number of iterations.

The constraints are given here, as presented in Equation (9).

$$\begin{cases} a(0) = 2, a(T) = 0, a\left(\frac{T}{2}\right) = 1 \\ a'(0) < 0, a'(T) < 0 \end{cases} \tag{9}$$

Thus, $a$ could be indicated by Equation (10).

$$a(t) = \frac{4+2r_3}{T^3}t^3 - \frac{6+3r_3}{T^2}t^2 + r_3\frac{t}{T} + 2 \\ -3.0 \times 10^{-3} < r_3 < 0 \tag{10}$$

Figure 1 shows the nonlinear convergence factor evolution at different values of $r_3$. The maximum number of iterations is set to 500. The three values of $r_3$ are $-2 \times 10^{-3}$, $-1 \times 10^{-3}$, and $-2 \times 10^{-4}$. The convergence factor values for the three curves decrease with the number of iterations. With the increase of iterations, the cut-off point between global and local search is reached when $T_0 = 250$. Global and local searches could be equally divided and effectively balanced in this condition. In addition, the curve corresponding to $r_3 = -2 \times 10^{-3}$ is lower than the other two curves at the early search stage and higher at the late search, indicating its focus on local search and adequate step size. Considering the drawback of converging to the local optimum in GWO, the curve corresponding to $r_3 = -2 \times 10^{-3}$ was chosen to ensure the property of jumping out of the local optimum in the study.

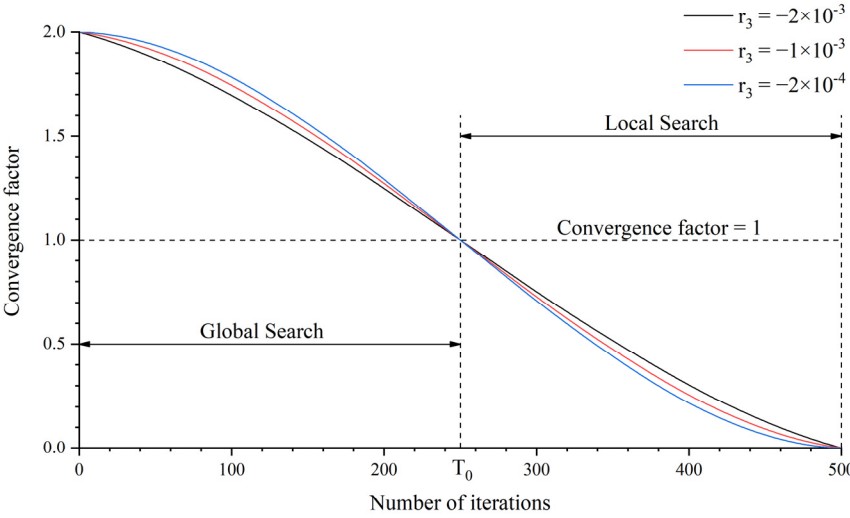

**Figure 1.** Nonlinear convergence factor based on third-degree polynomials.

### 2.2.3. Weighting Factors Based on Euclidean Norm and Hierarchy

In the GWO strategy, the position of the gray wolf is updated by the average formula. One limitation of this strategy is that the leadership of the wolf located in the first rank is not considered. Another weakness is that the weighting factors are kept constant as the iteration proceeds. Weighting factors based on Euclidean norms and hierarchy are proposed to overcome this problem, as shown in Equation (11).

$$\begin{cases} \rho_1 = \frac{\|X_1\|}{\|X_1\|+\|X_2\|+\|X_3\|} \\ \rho_2 = \frac{\|X_2\|}{\|X_1\|+\|X_2\|+\|X_3\|} \\ \rho_3 = \frac{\|X_3\|}{\|X_1\|+\|X_2\|+\|X_3\|} \end{cases} \tag{11}$$

Here $\rho_i$ $(i = 1, 2, 3)$ are the weighting factors of the first-three level wolves, respectively; $X_i$ $(i = 1, 2, 3)$ are the Euclidean norms of the first three levels, respectively.

The weights of the wolf $\alpha$, $\beta$, and $\gamma$ are multiplied by 0.6, 0.3, and 0.1 to reinforce the wolf pack hierarchy. The formula for updating the location of the gray wolf can be improved, as Equation (12) states.

$$X(t+1) = \frac{6\rho_1 \cdot X_1 + 3\rho_2 \cdot X_2 + \rho_3 \cdot X_3}{10} \tag{12}$$

where $\rho_i$ $(i = 1, 2, 3)$ are the weighting factors of the first-three level wolves, respectively; $X_i$ $(i = 1, 2, 3)$ represent the position vectors of the wolf $\alpha$, $\beta$, and $\gamma$ respectively.

### 2.3. Numerical Experiment of Algorithm Performance

A numerical experiment was performed to verify the effectiveness of IGWO. Six typical functions were selected for simulation in the experiments, including Sphere, Rosenbrock, Quartic, Rastrigin, Ackley, and Griewank. Table 1 shows the mathematical expressions, dimensions, and search ranges of these typical functions. Sphere, Rosenbrock, and Quartic are single-peak functions. Especially Quartic is a multidimensional flat bottom function with random disturbances. The single-peak functions are mainly applied to determine the accuracy of IGWO. Rastrigin, Ackley, and Griewank are multi-peaked functions that tend to cause the algorithm to converge to a locally optimal solution. The performance to jump out of the local optimum could be tested reasonably for IGWO. In addition, the experimental results of IGWO, SGWO [33], and GWO are compared in the simulation.

**Table 1.** Test functions in the numerical experiment.

| Test Function | Mathematical Expression | Dimension | Search Range |
|---|---|---|---|
| Sphere | $\sum\limits_{i=1}^{D} x_i^2$ | 30 | [−100, 100] |
| Rosenbrock | $\sum\limits_{i=1}^{D-1}\left[100\left(x_{i+1}-x_i^2\right)^2+\left(x_i-1\right)^2\right]$ | 30 | [−30, 30] |
| Quartic | $\sum\limits_{i=1}^{D} ix_i^4 + random[0, 1]$ | 30 | [−1.28, 1.28] |
| Rastrigin | $\sum\limits_{i=1}^{D-1}\left[x_i^2 - 10cos(2\pi x_i)+10\right]$ | 30 | [−5.12, 5.12] |
| Ackley | $-20exp\left(-0.2\sqrt{\frac{1}{30}\sum\limits_{i=1}^{D} x_i^2}\right) - exp\left(\frac{1}{30}\sum\limits_{i=1}^{D} cos2\pi x_i\right) + 20 + e$ | 30 | [−32, 32] |
| Griewank | $\frac{1}{4000}\sum\limits_{i=1}^{D} x_i^2 - \prod\limits_{i=1}^{D} cos\left(\frac{x_i}{\sqrt{i}}\right) + 1$ | 30 | [−600, 600] |

To ensure a fairness, the population size is 30, and the maximum number of iterations is 500 for IGWO, SGWO, and GWO. The three algorithms were performed 30 times independently for each function, and the average values were taken as the simulation results. The results of the numerical experiments are given in Table 2. The optimal values simulated by IGWO are closer to the theoretical optimal solution than the results of the other two algorithms. Among them, the simulation results of Sphere, Rastrigin, and Griewank are equal to the theoretical values, revealing the high accuracy of IGWO.

Figure 2 presents the convergence curves of the six test functions with IGWO, SGWO, and GWO. The convergence curves of all three algorithms continue to decrease as the number of iterations increases. In particular, the convergence curve of IGWO decreases significantly faster than the corresponding curves of the other two algorithms, indicating the progress of IGWO in terms of running time. Furthermore, for the multi-peaked test function, the convergence curve of IGWO continues to decrease while the search of the other two algorithms converges to a local optimum solution. This result demonstrates the ability to jump out of the local search and converge to the global optimum value in IGWO.

**Table 2.** Results of the numerical experiments.

| Test Function | Optimization Algorithm | Simulated Optimum Value | Theoretical Optimum Value |
|---|---|---|---|
| Sphere | IGWO | 0 | |
| | SGWO | $6.45 \times 10^{-33}$ | 0 |
| | GWO | $1.34 \times 10^{-26}$ | |
| Rosenbrock | IGWO | $2.89 \times 10$ | |
| | SGWO | $2.70 \times 10$ | 0 |
| | GWO | $2.72 \times 10$ | |
| Quartic | IGWO | $2.04 \times 10^{-4}$ | |
| | SGWO | $2.83 \times 10^{-4}$ | 0 |
| | GWO | $1.40 \times 10^{-3}$ | |
| Rastrigin | IGWO | 0 | |
| | SGWO | $5.68 \times 10^{-14}$ | 0 |
| | GWO | $1.71 \times 10^{-12}$ | |
| Ackley | IGWO | $4.44 \times 10^{-15}$ | |
| | SGWO | $1.51 \times 10^{-14}$ | 0 |
| | GWO | $1.11 \times 10^{-13}$ | |
| Griewank | IGWO | 0 | |
| | SGWO | $1.16 \times 10^{-2}$ | 0 |
| | GWO | $2.84 \times 10^{-2}$ | |

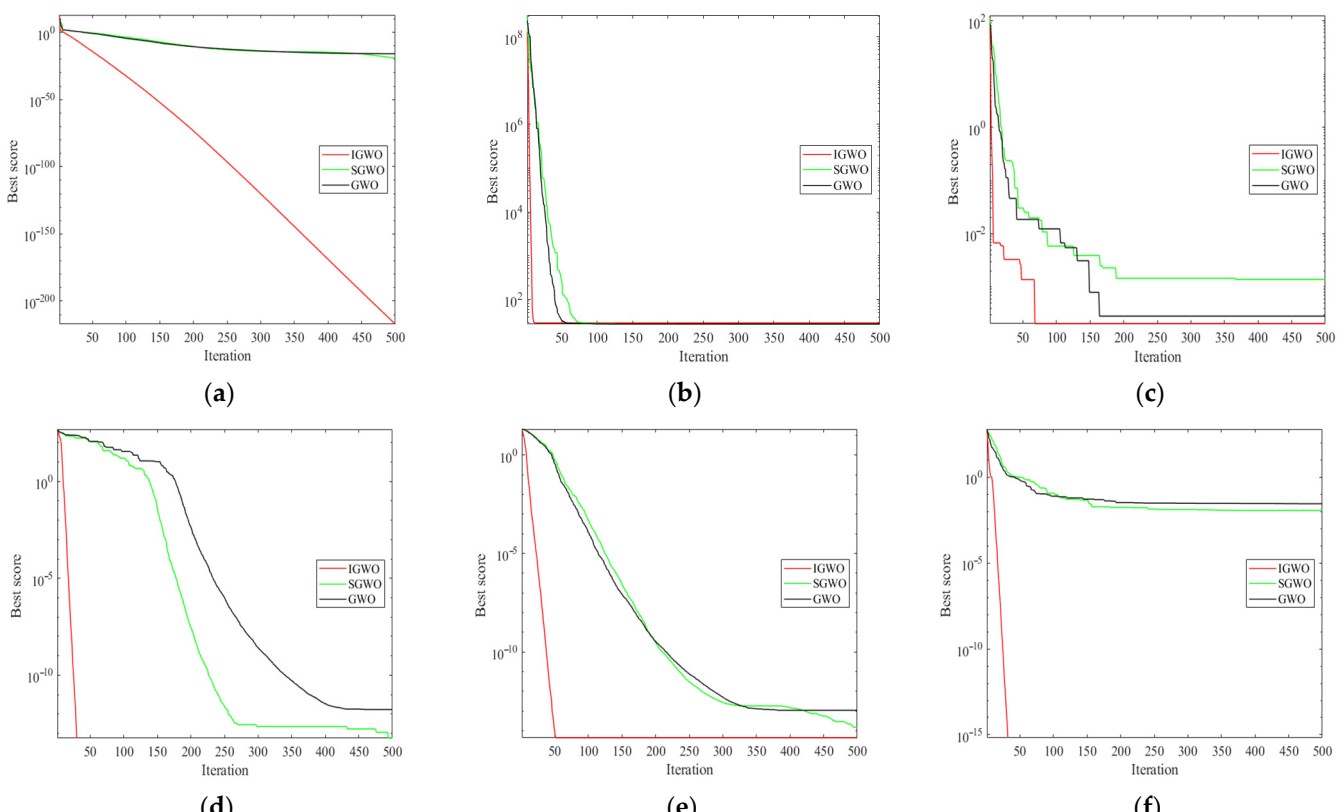

**Figure 2.** Convergence curves of the test function with the three optimization algorithms: (**a**) Sphere; (**b**) Rosenbrock; (**c**) Quartic; (**d**) Rastrigin; (**e**) Ackley; (**f**) Griewank.

## 3. Inverse Model of Seepage Parameters

### 3.1. The Objective Function

The aim of the inverse model is to determine the hydraulic conductivity for each partition while minimizing the value of the objective function. The objective function was constructed by hydraulic head and leakage in this paper, improving the reliability of the simulation results. Suppose that the hydraulic conductivity of each medium is isotropic. Denote by $K = [k_1, k_2, \ldots, k_n]$ the combination of hydraulic conductivity, in which $k_i$ represents the hydraulic conductivity of the $i$th media. The number of piezometers and measuring weirs are indicated by $m$ and $n$, respectively. $H = [H_1, H_2, \ldots, H_M]$ is expressed as a sequence of hydraulic head measurements. Similarly, $Q = [Q_1, Q_2, \ldots, Q_N]$ is a series of leakage volume measurements. $H_i(K)$ and $Q_j(K)$ are the simulated hydraulic head and leakage volume by finite element method. The mathematical model for the inverse problem is established, as shown in Equation (13).

$$minf = \left( \sum_{i=1}^{M} \frac{\|H_i(K)-H_i\|_2^2}{\|H_i\|_2^2} \right)^{\frac{1}{2}} + w \left( \sum_{i=1}^{N} \frac{\|Q_j(K)-Q_j\|_2^2}{\|Q_j\|_2^2} \right)^{\frac{1}{2}} \tag{13}$$
$$\text{s.t. } K_{min} \leq K \leq K_{max}$$

Here $K_{min}$ and $K_{max}$ are the lower and upper bounds of hydraulic conductivity values, respectively. The range of hydraulic conductivity can be roughly determined by geological data and engineering experience. $w$ is a weight factor for balancing the hydraulic head and the leakage volume. In this paper, the simulated leakage value is estimated by the flow rate and area of the overwater cross-section. It is suggested that the value of the weighting factor is set small considering an error between the simulated value and the measurement at the shoulder part of the dam. Zhou [23] compared the relative errors of hydraulic head and leakage volume at different weights. The results show that the simulated values are in good agreement with the measurements, and the minimum value of relative error is reached at the condition of $w = 0.02$. The finding was applied in this paper.

The objective combination of hydraulic conductivity was obtained by the searching process. The search process was accelerated by IGWO. The objective combination of hydraulic conductivity was applied to simulate the seepage field of the dam during operation.

### 3.2. Procedure of the Inversion Model

The procedure of the inversion model could be summarized by specific steps. The steps are as follows.

Step 1: Set initial parameters of IGWO. The number of search agents $S$, the maximum number of iterations $T$, and the bounds of hydraulic conductivity values, $K_{min}$ and $K_{max}$, are determined initially.

Step 2: Initialize the population. Equation (1) is applied for initialization, ensuring that the initial populations are uniformly distributed in the solution space.

Step 3: Calculate the fitness of individual gray wolves. The three gray wolves with the top fitness values are selected as $\alpha$, $\beta$, and $\gamma$.

Step 4: Update the position. Determine the distance of the gray wolf from the three dominant wolves, respectively. Orient the location and calculate the weighting factors of the three wolves. The position of the gray wolf at the fourth level is updated according to Equation (12).

Step 5: Iterative Judgment. Determines whether the maximum number of iterations has been achieved. If not, skip to the third step. Otherwise, end the iterative procedure. Output the objective hydraulic conductivity and the corresponding fitness value.

Step 6: Positive verification. The combination of the target hydraulic conductivity is substituted into the finite element model for positive verification. Compare the calculated and monitored values of hydraulic head and leakage volume and evaluate the reasonableness of the simulation.

Figure 3 presents the flow chart of the model.

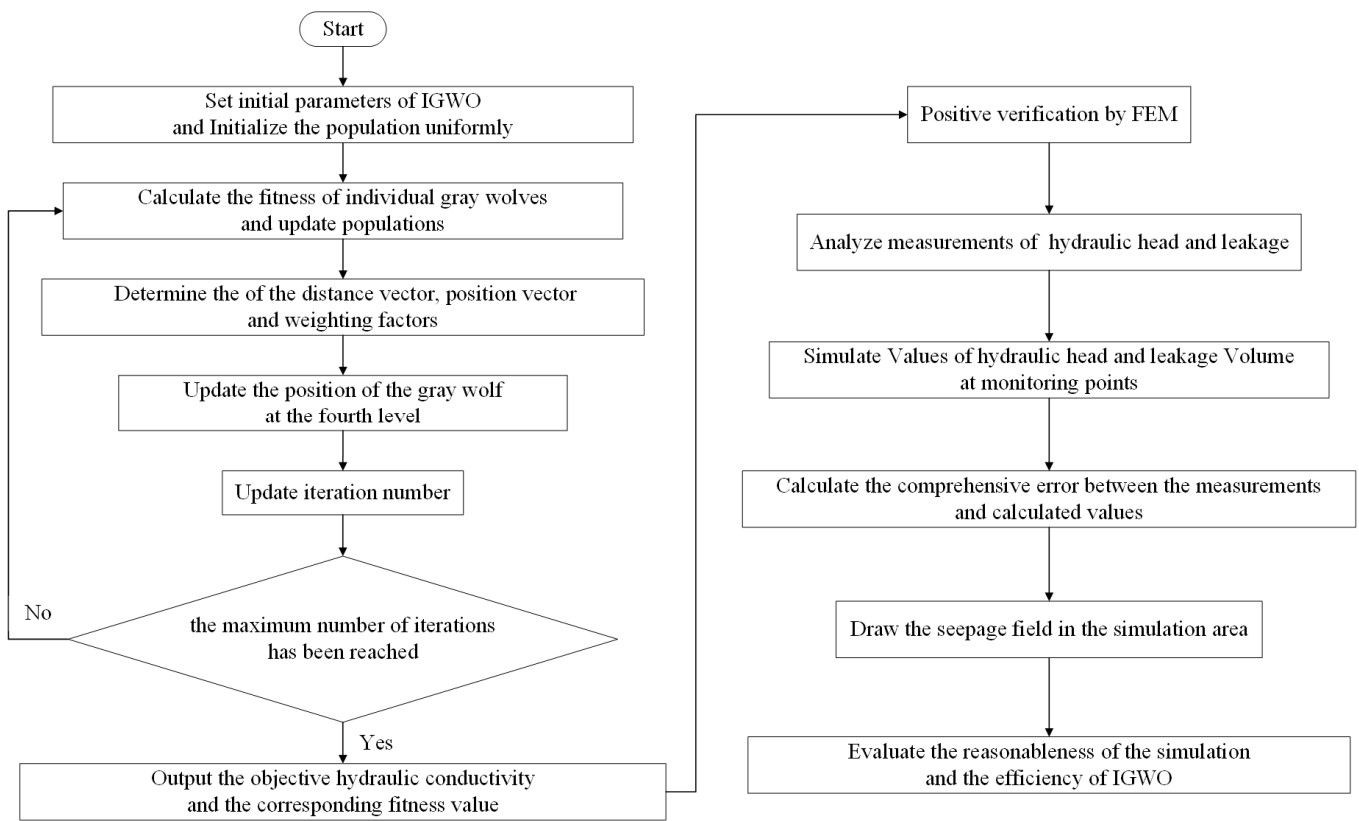

**Figure 3.** Flow chart of the inverse model based on IGWO.

## 4. A Casebook Study

### 4.1. Project Overview

Located on the Muli River in the Sichuan Province of China, the Kakiwa Hydropower Station is a project focused on power generation and ecological preservation. The location of the Kakiwa Dam is indicated in Figure 4.

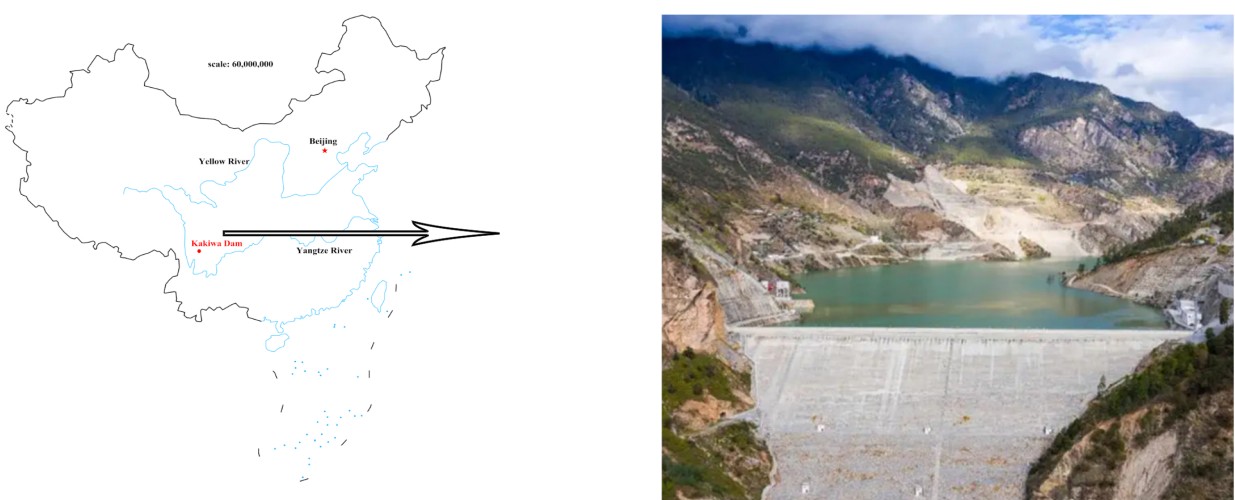

**Figure 4.** The location of the Kakiwa Dam.

A concrete panel rockfill dam is selected as the barrage in the pivot project, with a maximum height of 171 m. The crest width of the dam is 11 m. The dam mainly consists of a concrete face slab, blanket area, cushion area, transition area, rockfill area, drainage area, ballast area, and grout curtain. The normal storage level is 2850.00 m, the calibration flood

level is 2852.20 m, and the dead water level is 2800.00 m. Figure 5 shows the maximum cross-section of the dam body.

**Figure 5.** The max cross-section of the dam body.

A total of 37 piezometers were installed for seepage monitoring of the dam body and dam foundation. Among them, the piezometer $P_{DB-13}$ is located downstream of the curtain. $P_{DB-24} - P_{DB-27}$ are installed near the original ground line. Figure 5 presents the locations and elevations of these piezometers. The water measuring weir is installed at the downstream cofferdam axis of the dam.

The hydraulic conductivities of the five media, including the grout curtain, the top cover layer, the second cover layer, the moderately weathered zone, the slightly weathered zone, and the fresh bedrock zone, are limited to reasonable ranges and needed to be optimized. The parameter ranges are given in the results of the simulation. In addition, the hydraulic conductivities of other media are indicated in Table 3.

**Table 3.** Hydraulic conductivity of stationary medium.

| Material | Hydraulic Conductivity (m/s) |
|---|---|
| Upstream face slab | $1.00 \times 10^{-6}$ |
| Bedding material | $9.90 \times 10^{-4}$ |
| Transition material | $7.50 \times 10^{-2}$ |
| Main rock-fill zone | $8.70 \times 10^{-1}$ |
| Secondary rock-fill zone | $9.80 \times 10^{-1}$ |
| Upstream blanket | $1.00 \times 10^{-5}$ |

*4.2. Analysis of Monitoring Data*

Figure 6 shows the monitoring data of the piezometers around the grout curtain and the water measuring weirs during the operation period. The hydraulic head on the downstream side of the curtain is relatively consistent with the upstream water level. The hydraulic head measured by the piezometer lags behind the upstream head, which is called the hysteresis effect. The measured hydraulic head rises less than the upstream water level. In addition, the value of the piezometer $P_{DB-24}$ is approximately 2776 m, and the difference among the values of piezometers $P_{DB-24}$ to $P_{DB-27}$ is not significant, indicating the efficiency of the impermeable curtain.

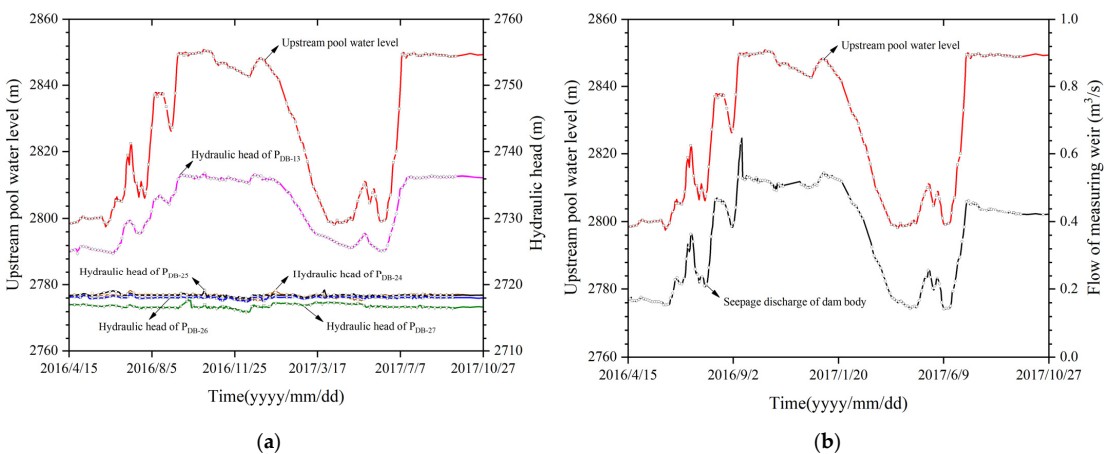

**Figure 6.** Measurements of hydraulic head and seepage discharge during storage period: (**a**) hydraulic head; (**b**) seepage volume.

Similarly, the seepage volume is consistent with the upstream water level. The value of seepage volume at the dam body is relatively small, with a stable value of 0.41 m$^3$/s in the operation period.

It is assumed that the seepage field is stable for simplicity. A period with a slight variation of upstream and downstream reservoir levels and long duration was selected for the inverse model, which could minimize the seepage lag effect to a certain extent. As seen in Figure 6, the upstream pool level is relatively stable from 10 July 2017, to 27 October 2017, with values between 2849.55 m and 2850.06 m. Therefore, this period was chosen for the inversion model.

*4.3. Computation Model*

IGWO was used for iterative search. The maximum number of iterations is 200, and the initial population size is 30. The simulation is performed 20 times independently by IGWO, and the average values of simulated hydraulic conductivity are taken as the target results.

The multi-physics field simulation software COMSOL Multiphysics was applied in this simulation. A two-dimensional finite element mesh was created for inverse modeling each medium's hydraulic conductivity. The seepage properties of the dam body and foundation were analyzed. The free tetrahedral grid is applied to build the mesh, composed of 11,635 domain elements and 14,400 vertices. The maximum and minimum widths of the cells are 20 m and 5 m, respectively. A right-hand cartesian coordinate system is constructed with the *x*-axis pointing to the downstream reservoir and the *z*-axis pointing to the sky vertically. The upstream and downstream are both extended by 2.5 times the dam's height on the *x*-axis. The depth of the foundation is taken as 400 m. In addition, the upstream and downstream water levels are, respectively, set at 2850 m and 2702 m. The two-dimensional finite element mesh is presented in Figure 7.

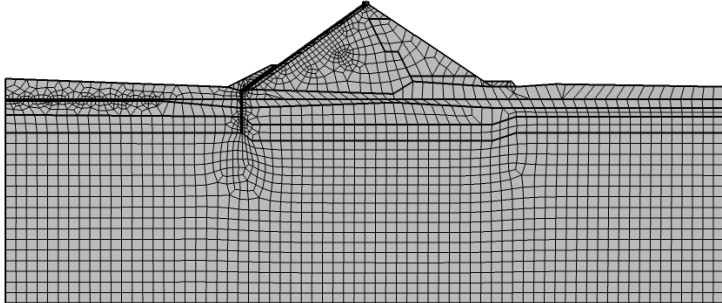

**Figure 7.** 2D FE mesh of Kakiwa dam.

### 4.4. Results of the Simulation

#### 4.4.1. Hydraulic Conductivity

The hydraulic conductivity of each medium was determined by IGWO. The hydraulic conductivity of each medium at the dam site is given in Table 4. All the results are within the corresponding search range.

**Table 4.** Hydraulic conductivity of each medium determined by IGWO.

| Material | Hydraulic Conductivity (m/s) | Search Range (m/s) |
|---|---|---|
| Grout curtain | $3.00 \times 10^{-5}$ | $1.00 \times 10^{-6}$–$1.00 \times 10^{-4}$ |
| The top cover layer | $5.33 \times 10^{-2}$ | $1.00 \times 10^{-3}$–$1.00 \times 10^{-1}$ |
| The second cover layer | $2.67 \times 10^{-3}$ | $1.00 \times 10^{-4}$–$1.00 \times 10^{-2}$ |
| Moderately-weathered zone | $5.50 \times 10^{-4}$ | $1.00 \times 10^{-5}$–$1.00 \times 10^{-3}$ |
| Slightly weathered zone | $1.10 \times 10^{-4}$ | $1.00 \times 10^{-5}$–$1.00 \times 10^{-3}$ |
| Fresh bedrock zone | $3.10 \times 10^{-5}$ | $1.00 \times 10^{-6}$–$1.00 \times 10^{-4}$ |

#### 4.4.2. Hydraulic Head

The results of hydraulic conductivity were substituted into the finite element model for positive analysis to verify the reasonableness of this simulation. The simulated values of the hydraulic head and leakage volume at the monitoring points were obtained and compared with the corresponding measurements. Absolute and relative errors of hydraulic head and leakage were calculated. A contour of the hydraulic head in the dam site area was also predicted.

The mathematical expression for the relative error of the hydraulic head is given in Equation (14):

$$\delta_H = \frac{|H_i - H|}{\Delta H} \times 100\% \tag{14}$$

where $\delta_H$ is the relative error of the hydraulic head; $H_i$ and $H$ represent the simulated and measured values of the hydraulic head, respectively. $\Delta H$ denotes the difference in water level between the upstream and downstream sides, taken as 148 m.

Table 5 compares the calculated and measured hydraulic head values at the monitoring points. The calculated values at the monitoring points are relatively close to the measurements. Among them, the simulated values at the piezometers $P_{DB-24} - P_{DB-27}$ show a very high consistency with the corresponding measurements. The maximum value of the absolute error $-1.41$ m, and the maximum value of the relative error is 0.95%. Meanwhile, the value of the hydraulic head decreases with the increase of seepage distance.

**Table 5.** Comparison between the calculated and measured hydraulic head values.

| Monitoring Points | Measured Hydraulic Head (m) | Simulated Hydraulic Head (m) | Absolute Error (m) | Relative Error (%) |
|---|---|---|---|---|
| $P_{DB-13}$ | 2736.17 | 2735.74 | −0.43 | 0.29 |
| $P_{DB-24}$ | 2718.67 | 2717.51 | −1.16 | 0.78 |
| $P_{DB-25}$ | 2718.28 | 2717.16 | −1.12 | 0.76 |
| $P_{DB-26}$ | 2718.21 | 2716.80 | −1.41 | 0.95 |
| $P_{DB-27}$ | 2716.61 | 2716.24 | −0.37 | 0.25 |

#### 4.4.3. Leakage Volume of the Dam Foundation

The seepage volume is estimated by the flow velocity and the overflow surface. The formula for calculating leakage volume is given in Equation (15).

$$Q_i = \iint vBdxdz \tag{15}$$

where $Q_i$ is the simulated seepage volume; $v$ means the flow rate at the vertical spillway surface; $B$ represents the length of the dam taken as 355 m.

The relative error for the leakage volume is determined by Equation (16).

$$\delta_Q = \frac{|Q_i - Q|}{Q} \times 100\% \tag{16}$$

where $\delta_Q$ is the relative error of the dam body leakage; $Q_i$ and $Q$ are the simulated and measured values of the dam body leakage, respectively.

Table 6 shows the comparison of the calculated and measured values of the dam body leakage during the stable upstream water level. The simulated leakage values are in good agreement with the actual measurements, showing the accuracy of the IGWO strategy and the reliability of the simulation.

**Table 6.** Comparison between the simulated and measured values of the dam body leakage.

| Leakage | Measured Values (m$^3$/s) | Simulated Values (m$^3$/s) | Absolute Error (m$^3$/s) | Relative Error (%) |
|---|---|---|---|---|
| Dam body | 0.40 | 0.38 | −0.02 | 5.00 |

The leakage measurements are averaged over the simulation period. The relative error of the calculated seepage volume is 5.00%, demonstrating the positive performance of IGWO in the simulation.

Figure 8 presents the contour map of the hydraulic head at the dam site. The distribution of contours is in accordance with seepage characteristics. The results show the accuracy of the simulation and the reasonableness of the calculated hydraulic conductivity.

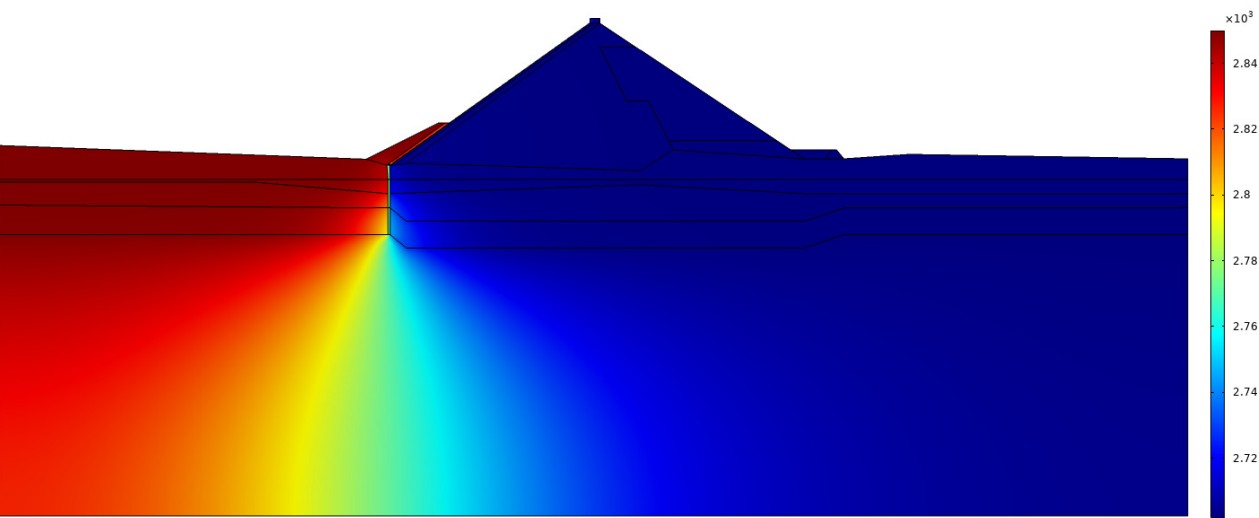

**Figure 8.** Distribution of seepage contours at the maximum section of the dam.

## 5. Discussion

The hydraulic conductivities of the dam and dam foundation change with operation and loading, affecting the effectiveness and safety of the hydraulic project. An Improved Gray Wolf Optimizer for solving the problem that hydraulic conductivity is not easily determined, is introduced in this paper.

The objective hydraulic conductivity was determined based on IGWO. The obtained hydraulic conductivities of the dam and dam foundation were applied in the finite element positive analysis. The results show the effectiveness of IGWO in determining hydraulic conductivity. It is suggested that IGWO could be used to obtain reasonable simulation results in similar inverse problems.

The hydraulic conductivity of each medium in the dam body and dam foundation simulated in this paper was obtained with a stable upstream water level. In fact, the upstream water level of a reservoir varies continuously with the operating conditions and purpose. The hydraulic conductivity will also change with the monitored values. It is worth noting that IGWO is still applicable under this condition.

## 6. Conclusions

Aiming at balancing the local and global search of GWO, along with uniformity and stochasticity, IGWO was proposed in this paper. The improvement of IGWO in accuracy and running time was indicated in numerical experiments. IGWO was used in the inverse modeling of the hydraulic conductivity of the Kakiwa dam. The hydraulic head and leakage were used to set up the objective function. Errors in the hydraulic head and leakage were calculated. The steady seepage field was analyzed in the application case. The main conclusions of this paper are given as follows.

(1) The performance of IGWO is improved due to the three strategies. A novel approach for initialization contributes to populations of semi-uniform and semi-random. The polynomial-based nonlinear convergence factor is selected to keep the equilibrium of the local and global search. Weighting factors based on Euclidean norms and hierarchy helps to update the position of the wolf dynamically.

(2) A numerical experiment was conducted to demonstrate the performance of IGWO. The optimal values obtained by IGWO are closer to the theoretical solution than the results of the other two algorithms. The results of the experiment demonstrate the feasibility and efficiency of IGWO.

(3) The target combination of hydraulic conductivity was obtained by IGWO. The values of hydraulic conductivities were substituted into the finite element model. The values of the hydraulic head and leakage quantity at the corresponding measurement points were obtained. The maximum values of absolute and relative errors of the hydraulic head were—1.41 m and 0.95%, respectively. The absolute and relative errors of the seepage volume were—0.02 $m^3$/s and 5.00%, respectively. The results of the application case show that the inversion model and the algorithm are reliable and efficient.

**Author Contributions:** Conceptualization, Y.S. and L.X.; methodology, Z.S.; software, Y.S.; validation, J.D. and L.J.; formal analysis, L.X.; investigation, J.D.; resources, Z.S.; data curation, L.J. and Q.L.; writing—original draft preparation, Y.S.; writing—review and editing, Y.S.; visualization, J.D. and Q.L.; supervision, Z.S.; project administration, Z.S.; funding acquisition, L.X. All authors have read and agreed to the published version of the manuscript.

**Funding:** This research was funded by the "National Key R&D Program of China, grant number 2019YFC1510802".

**Institutional Review Board Statement:** Not applicable.

**Informed Consent Statement:** Not applicable.

**Data Availability Statement:** Not applicable.

**Conflicts of Interest:** The authors declare no conflict of interest.

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
