# Peer review of "Inverse Modeling of Seepage Parameters Based on an Improved Gray Wolf Optimizer"

_applsci, doi:10.3390/app12178519_

Round 1
Reviewer 1 Report
This paper concentrates on solving an inverse seepage model based on field measured data. Generally, the authors address the problem in appropriate method. However, there are many sections need to be revised and more clarifications are required. Also, the English writing style is good, but there are many defects could be noticed in some places of manuscript. physically, the researchers needs to focus more on the physical issue and simulation model as much as on the optimization model because it is the major part of the problem.

Reviewer 2 Report
General Comment:
Based on the inverse analysis, Gray Wolf Optimizer (GWO) is introduced into this study to search the target hydraulic conductivities. Three strategies are proposed to improve the performance of the algorithm. The performance of the Improved Gray Wolf Optimizer (IGWO) is verified in the numerical experiments. The results show the feasibility of Improved Gray Wolf Optimizer (IGWO) in determining the seepage parameters of the dam. The obtained predictions are presented and discussed. From the obtained results, the authors conclude about the reliability of the proposed model. The article is written in an acceptable manner and has valuable content for the Journal's audience. It is a good example of applications of optimization method. Presentations and discussion are well presented.
The theme of the manuscript is innovative and reflects the application of intelligent algorithms in engineering. The submitted manuscript constitutes a good contribution to the field and can also serve as a guide for future research.
Specific Comment:
1) In the INTRODUCTION part, the application of optimization algorithms in different fields is indicated. However, relevant research background needs to be supplemented, especially the application of algorithms in the field of seepage. And you should cite the paper all you used properly.
2) The case study should be better described. The hydraulic conductivity of each layer medium used in the inversion model is given in this paper. Other calculation parameters (such as hydraulic conductivity not used for inversion) should be added in the application case.
3) In general, there is a lack of explanation of replicates and statistical methods used in the study. For example, the authors need to clarify this question: Why use a function based on a third-degree polynomial to fit convergence factors in the paper? Why not polynomials of some other degree?
4) In Figure 5, it appears that both the upstream water level and measurements of hydraulic head are stable from October 2016 to December 2016. Why is this period not used in the inversion model? Hope to see the explanation of the difference between this period and the other.
5) In the project overview, thirty-seven piezometers are installed for seepage monitoring. However, only the measurements of five piezometers are used in the inversion model. Please explain the reasonableness of this treatment.
6) I read the results and discussion section completely. The discussion section is the main part of a paper, but this manuscript mainly reported the data of the modeling without discussing it through adding available reasoning for justifying the result. I recommend author adding several reasoning and comparison through available publications in the literature.
7) In the application case, there is no information about the number of replicates and experimental design. Further clarification of the content and additional experimental details are requested.
8) Both figures and tables must come immediately after the first time it is mentioned in the text, not far away, e.g. Figure 3.
9) Eliminate the use of redundant words, e.g. in this way, therefore, in addition, therefore, this way. In fact, sometimes there is no cause and effect or transitions between two sentences. Revise all similar cases, as removing these term(s) would not significantly affect the meaning of the sentence.
Besides, a few errors should be corrected.
10) 55 - “determinine” – “determine”
11) 130 and 131 - “, , and respectively” – comma missing
12) 205 - Table 4. – “[-100,100], [-30,30], [-1.28,1.28], [-5.12,5.12], [-32,32], [-600,600] and [0,1]” spacing missing
13) 248 - “w=0.02” – spacing missing
14) 256 - “presentes” – “presents”
15) 325 - “was” – is
16) 371 - “5.00 %” – extra space
17) 410 and 411 - “0.95 %” and “5.00 %” – extra space
Round 2
Reviewer 1 Report
A quick revision for English writing stile and this paper will be ready for publishing.